# QUADCAL: CALIBRATION FOR IN-CONTEXT LEARNING

## ABSTRACT

Large language models (LLMs) are increasingly being applied to high-stakes domains with high consequences for errors such as healthcare, drug discovery, law, and finance. However, they are often unstable and highly sensitive to prompt design, which can introduce contextual bias into their predictions. To mitigate this bias, various calibration methods have been developed to prevent overconfident and incorrect predictions. Existing techniques are either confidence-based, relying on heuristics to quantify bias, or likelihood-based, which is theoretically grounded but introduces unnecessary computational overhead. In this work, we introduce **QuadCal**, a novel supervised likelihood-based calibration method that is up to 40% faster and outperforms the existing likelihood-based approach. Specifically, QuadCal leverages Quadratic Discriminant Analysis (QDA), a supervised algorithm that directly models class-conditioned distributions, making it more efficient. We evaluated calibration methods on GPT-2 models and the more recent Llama and Gemma's instruction-tuned (IT) models, which are harder to calibrate. Empirically, we show that on average over seven different natural language classification datasets, QuadCal outperforms existing methods on GPT-2 models and is competitive with earlier methods on IT models.

## 1 INTRODUCTION

Large language models (LLMs) have demonstrated remarkable performance across a wide range of classification tasks and domains. They are increasingly being adopted for many critical domains such as healthcare, drug discovery, law, and finance (Naveed et al., 2023). The consequences of wrong predictions in such high-stakes domains are very high, ranging from severe financial losses and wrong judgements to clinical misdiagnoses. Therefore, it is essential to ensure the reliability and trustworthiness of the LLM that are used in these domains.

A major breakthrough in improving the adaptability of LLMs has come from the observation of a specific ability of LLMs known as **in-context learning** (ICL) (Brown et al., 2020). ICL enables LLMs to perform new tasks by conditioning the pre-trained LLM on a text input containing a few examples or instructions for the new task and then generating the next tokens as prediction. Notably, ICL does not require parameter updates and learns only via the input prompts. This makes ICL a great choice for adaptation to new domains where fine-tuning is expensive.

However, LLMs are often observed to be poorly calibrated (Chen et al., 2022), making them unreliable in automated systems or use in critical domains. A poorly calibrated model will provide overconfident or underconfident predictions which will result in serious consequences in such domains. Overconfidence is particularly severe in LLMs, and it has been observed that they tend to 'hallucinate' (Huang et al., 2025), that is, they provide highly confident but factually wrong answers. This misleads the user and makes it difficult to rely on the model's output. Similarly, underconfidence can also be equally misleading and reduces the reliability of the downstream decision making systems. For example, a poorly calibrated model might predict a chemical compound to be non-toxic with a 90% confidence score. Ideally, 90% of such predictions would be safe, but in reality, only 50% may turn out to be safe and the rest toxic. Conversely, underconfident scores will result in discarding potential non-toxic compounds for further testing. Such miscalibrated predictions might result in using those harmful compounds for further testing costing money and risking human life.

Furthermore, in ICL, the predictions are made solely based on the prompt input, which we denote as the **context** $C$. The model estimates the conditional probability $P(y|C)$, where $y$ is the predicted output. This makes ICL vulnerable to **contextual bias**, where the format or examples of a prompt and its ordering could cause instability due to variance in prediction (Zhao et al., 2021). Recently, OptiSeq (Bhope et al., 2025) was proposed as a method for selecting the optimal ordering of examples for ICL. However, it requires us to evaluate all permutations of the ordering of the examples, which can be computationally expensive and difficult to apply for large example sets. ICL is also sensitive to prompt formatting, despite the prompts having the same intended meaning (He et al., 2024). Subtle changes in the prompt, such as adding a white space or punctuation, can also cause instability (Seleznyov et al., 2025).

To mitigate these challenges, various calibration techniques have been developed specifically for ICL to handle contextual bias. The calibration methods for ICL broadly fall into two categories: **confidence-based** calibration and **likelihood-based** calibration. The confidence-based calibration methods estimate the model's bias and rescale the confidence scores so that they better align with the true probabilities. These methods are usually simple and easy to implement but are based on heuristics to compute contextual bias. The likelihood-based calibration methods take a probabilistic Bayesian approach by explicitly modeling the class-conditioned distributions of the model's outputs. The Bayes' theorem plays a fundamental role in probability theory (Bishop & Nasrabadi, 2006). In general, Bayesian approaches are preferred because they provide a principled framework that takes prior knowledge into consideration and updates the posterior distribution accordingly. The likelihood-based calibration methods focus on improving prediction accuracy by modeling the underlying class distributions. Although these calibration methods can be computationally intensive than confidence-based methods, their Bayesian approach makes them more reliable and theoretically grounded.

We introduce **QuadCal**, a new calibration method which falls into the second category where we model the class-conditioned distribution directly to improve the accuracy and, in turn the robustness of the predictions. QuadCal takes a Gaussian approach for calibration, where we estimate the probability density of the model's outputs for each class and use it to make class predictions. This improves reliability for high-stakes applications without altering the underlying confidence scores. In addition, we also systematically evaluate existing calibration methods for ICL on various pretrained LLMs.

The main contributions of this paper are as follows:

- We propose **QuadCal** – a supervised alternative to the existing likelihood-based approach for ICL calibration.

- We provide a **systematic evaluation** of existing calibration methods for ICL on recent LLMs across diverse natural language tasks.

- Experiments show that **QuadCal consistently matches or outperforms** existing state-of-the-art (SOTA) calibration methods for ICL.

- QuadCal is consistently **faster** than the existing likelihood-based approach across models, shot settings, and datasets.

## 2 RELATED WORK

One of the earliest influential works on calibration for neural networks was by Guo et al. (2017), who observed that although deep neural networks significantly improved performance compared to shallow networks, they are often poorly calibrated. To address this, they introduced *temperature scaling*, which is a widely used post-processing calibration method. They also provided an overview of several calibration assessment metrics, including the reliability diagram, expected calibration error (ECE), maximum calibration error (MCE), and negative log likelihood (NLL).

As pre-trained LLMs and ICL became more prevalent, new challenges emerged. One such challenge is contextual bias, where model predictions can be heavily influenced by the prompt design. This necessitated the development of calibration methods specifically for ICL. In general, model calibration can be performed either during training or post training of a model. With the introduction of

many pre-trained LLMs and their ability to perform new tasks without any gradient updates through ICL, post-training methods become the natural choice.

One of the first calibration methods focused on ICL was introduced by Zhao et al. (2021) called as *contextual calibration* (**CC**). It uses content-free test inputs such as "N/A" to estimate the model's inherent bias toward or against each of the classes, which could then be used to rescale the confidence scores for real inputs. Following this, Fei et al. (2023) proposed *domain calibration* (**DC**) which uses random in-domain words instead of content-free test inputs to handle domain-label bias. Here, the domain-label bias is defined as the distance between the model's prior predictions with random English words and predictions with random in-domain words. The confidence scores for real inputs are then rescaled as in CC. More recently, Zhou et al. (2023) have proposed batch calibration (**BC**) to address contextual bias by using the input examples (batch) itself instead of content-free tokens or in-domain words. Here, bias is calculated by taking a mean of the predicted probabilities for each of the classes in that batch followed by rescaling the confidence scores for real inputs. All these methods fall into the first category of calibration methods for ICL, where we estimate the bias and rescale the confidence scores to better align with true probabilities.

A more theoretically grounded approach for calibration was proposed by Han et al. (2022) with the introduction of prototypical calibration (**ProCa**) which estimates prototypical clusters for each of the labels. When a new input is provided, the calibration is done by estimating the likelihood of it belonging to each of the prototypical clusters. ProCa falls in the second category of calibration methods for ICL where the likelihood is estimate and the decision boundary is shifted to improve prediction accuracy. Although CC, BC and DC are easy to implement and effective, Bayesian approaches are more theoretically sound since it explicitly estimates the class-conditional probabilities and calibrates the output based on likelihood.

## 3 QuadCal: Bayesian Calibration with QDA

### 3.1 Background

Motivated by the insights discussed above, we introduce **QuadCal** - a Gaussian approach to calibration that is faster and more efficient than the existing Gaussian-based calibration method, ProCa. In ProCa, prototypical clusters are built in a two-step process:

- A Gaussian Mixture Model (GMM) is first trained on a small random subset of samples to build $n$ clusters, where $n$ is the number of classes.

- Once the clusters are built, they are mapped to the $n$ classes using Munkres (Hungarian) algorithm (Kuhn, 1955).

One of the shortcomings of ProCa is that it relies on GMM – an unsupervised clustering algorithm that requires the computationally expensive Munkres algorithm to map the $n$ clusters to $n$ labels. This step is avoidable in a supervised setting, where the ground-truth labels are readily available. Although the Munkres algorithm is optimal, it is computationally expensive for multiclass settings with a complexity of $O(n^3)$, where $n$ is the number of classes. Moreover, GMM inherently uses the iterative Expectation-Maximization (EM) algorithm (Dempster et al., 1977) to estimate the parameters of the Gaussian components, further adding to the computational overhead.

To address these limitations, we propose **QuadCal**, a supervised Bayesian approach to calibration that directly models the class-conditioned distribution of the data, thus avoiding both the iterative GMM procedure and the post-hoc cluster-to-label mapping required in ProCa. QuadCal uses Quadratic Discriminant Analysis (QDA) (Hastie et al., 2009), a supervised classification method that models each class as a multivariate Gaussian distribution with its own mean and covariance. Figure 1 illustrates how QDA models two classes with distinct means and covariances, separated by a quadratic decision boundary.

Unlike Linear Discriminant Analysis (LDA), which assumes equal covariance across all classes, making it suitable only for homoscedastic data, QDA makes a more relaxed assumption. QDA allows each class to have its own covariance, making it well-suited for heterogeneous, real-world datasets. Each class is modeled as a multivariate Gaussian with its own mean $\mu_k$ and covariance $\Sigma_k$

of class $k$:

$$P(X|y = k) = \frac{1}{(2\pi)^{\frac{d}{2}}|\Sigma_k|^{\frac{1}{2}}} exp(-\frac{1}{2}(X - \mu_k)^T\Sigma_k^{-1}(X - \mu_k))$$

where $X \in \mathbb{R}^d$ and $d$ is the number of features of $X$. Once $\mu_k$ and $\Sigma_k$ are estimated, the quadratic discriminant function is given as:

$$\delta_k(x) = -\frac{1}{2}log|\Sigma_k| - \frac{1}{2}(x - \mu_k)^T\Sigma_k^{-1}(x - \mu_k) + log\pi_k$$

where $\pi_k$ is the prior probability of class $k$, and the classification is then given by:

$$\hat{y} = \arg\max_k \delta_k(x)$$

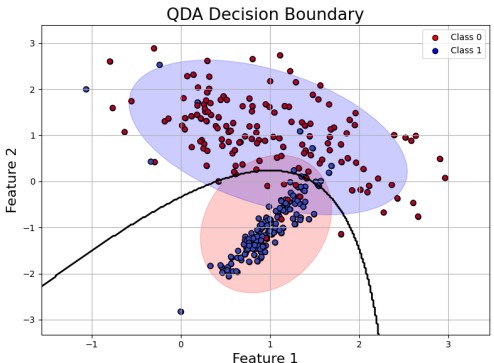

From a theoretical perspective, GMM could model more complex and multi-modal class distributions because it assumes that each class may consist of a mixture of multiple Gaussians, potentially making it more effective than QDA, which assumes a single Gaussian per class. However, ProCa enforces a restriction of exactly $n$ clusters, where $n$ is the number of classes, reducing the inherent flexibility of GMM. By using QDA, which estimates the Gaussian parameters directly, Quad-Cal avoids the computational overhead in ProCa while being functionally similar.

Figure 1: Illustration of class distributions modeled by QDA. Each class is a Gaussian with its own mean and covariance.

### 3.2 METHOD

To train the QDA model for QuadCal, we first construct an estimate set via stratified sampling of the training set of the target task for ICL, so that each class is well-represented. More details on the construction of the estimate set are provided in Section 4. The estimate set is then provided as input to a pre-trained LLM to obtain the probability outputs for each class label. We then apply a log transformation to these probabilities for numerical stability and to satisfy the Gaussian assumptions, following the approach used in ProCa. We then model the class-conditioned distribution $P(X|y = k)$, where $X \in \mathbb{R}^n$ is the log probability vector over $n$ classes, and $k \in \{1, ..., n\}$ represents the classes. For each class, the mean $\mu_k$ and covariance $\Sigma_k$ is estimated directly from the log-probabilities and the class prior is computed from the estimate set. Finally, for classification, each sample is assigned to the class with the highest discriminant score computed using the QDA model.

## 4 EXPERIMENTAL SETUP

**General setup:**
We largely follow the experimental setup of ProCa for a fair comparison. We evaluate QuadCal across a diverse set of natural language tasks such as sentiment, topic and entailment classification. The datasets considered are SST-2 (2 classes) (Socher et al., 2013), SST-5 (5 classes) (Socher et al., 2013), MR (2 classes) (Pang & Lee, 2005), all of which are sentiment classification tasks; Subj (2 classes) (Pang & Lee, 2004), a subjectivity classification task; AGNews (4 classes) (Zhang et al., 2015) which is a news topic classification task; RTE (2 classes) (Dagan et al., 2005), a textual entailment task; and TREC (6 classes) (Voorhees & Tice, 2000) which is a question classification task. The Amazon Polarity dataset (Zhang et al., 2015) was excluded due to computational constraints. We use the same prompt formats as in the original setup.

We included GPT-2-Large (0.8B) (Radford et al., 2019) and GPT-2-XL (1.5B) (Radford et al., 2019) from OpenAI to allow fair comparison with ProCa. It has been observed that instruction-tuned models are particularly difficult to calibrate (Zhu et al., 2023) and hence we chose two recent instruction-tuned models from both the Google (Team et al., 2025) and Meta (Grattafiori et al., 2024) families.

Among them, we picked those with comparable sizes to GPT-2 models to allow for fair comparison: Llama-3.2-IT (1B), Llama-3.2-IT (3B), Gemma-3-IT (1B) and Gemma-3-IT (4B).

We evaluated all models under 0-shot, 1-shot, 4-shot and 8-shot ICL settings. We compare QuadCal with three other calibration methods discussed in Section 2 – CC, BC and ProCa. Each experiment was repeated with five random seeds and the model performance was measured using classification accuracy. For all the datasets, the full test set was used for evaluation, except for AGNews, for which we randomly sampled 2000 examples.

**Estimate set construction:**
For both ProCa and QuadCal, we use stratified sampling instead of random sampling as used in ProCa. ProCa's GMM-based approach always generates $n$ clusters even with the estimate set having only representation from $n - 1$ classes. This is problematic since it completely and silently misses the underrepresented class, leading to incorrect cluster-to-class mapping during the Munkres step. This is particularly an issue for smaller datasets like RTE and TREC and reduces the quality of calibration. By using stratified sampling, every class is well represented, leading to more reliable prototypical clusters in ProCa and better class-conditioned distributions in QuadCal. The estimate set size is fixed as 100.

**Runtime analysis:**
To empirically evaluate the computational efficiency of QuadCal relative to ProCa, we designed a small experimental setup comparing the two methods across three datasets- SST-2 (2 classes), AGNews (4 classes), and TREC (6 classes) under 0-shot, 4-shot and 8-shot settings. This allows us to assess run time across varying numbers of classes and ICL shots. We chose to evaluate the larger models within each family for this analysis, and all the experiments were run with three random seeds.

For both methods, we report the end-to-end run time, including both the time required to train the GMM + Munkres (for ProCa) or QDA (for QuadCal) models and their inference time taken for calibration. It is to be noted that training GMM + Munkres or QDA models is a one-time cost, and if pre-trained models are available, the run time required for future evaluations would be further reduced. Moreover, when the output probabilities for the estimate set is already computed, both the training and inference for ProCa or QuadCal can be executed entirely on CPU. Nevertheless, we make a relative comparison here under identical experimental conditions.

All experiments for this analysis were run on a single node of a cluster using HTCondor. To ensure that no other jobs influenced the run time, we exclusively requested for all GPUs of the node, along with 5 CPU cores and 20 GB of system memory per job. The node is equipped with 4x A100 (40GB) and 512 GB of RAM.

**Significance testing:**
We performed significance testing to determine when QuadCal or ProCa is truly better than the other and not due to random chance. For each combination of model, shot, and dataset, we performed statistical tests on the accuracies obtained across five random seeds. We used a paired t-test to assess whether the mean difference in accuracy was significant for the two methods. Additionally, even if the mean difference is small, to check if a method is consistently better than the other, we did a binomial test. To consider both the magnitude and the direction of differences, for our analysis, we considered a result to be significant if it was significant in either the paired t-test or the binomial test. When the results are significant, it indicates either a higher mean accuracy or it consistently performs better than the other across the different runs. The null hypothesis for both tests is that there is no difference between the two methods and the significance was determined at $\alpha = 0.05$.

## 5 RESULTS

### 5.1 OVERALL PERFORMANCE

An overview of the average performance (macro-average accuracy) of the calibration methods across the considered pre-trained LLMs and various ICL shot settings is provided in Table 1. On average, QuadCal consistently matches or outperforms the other calibration methods, showing its effectiveness in improving test accuracy. In particular, **QuadCal achieves the highest average accuracy for all shot settings for the GPT-2 models and for Gemma-3-1B-IT**. Across all models, **CC gen-**

Table 1: Macro-average accuracy (%) of ICL (uncalibrated) and calibration methods (CC, BC, ProCa, QuadCal) across different pre-trained LLMs and ICL shot settings. First best results are in **bold** and second best are underlined. The full dataset-specific accuracies used to compute the macro-average, as well as macro-median accuracy (%), are reported in Appendix A.

| Model | Shots | Macro-average accuracy (%) | | | | |
|---|---|---|---|---|---|---|
| | | **ICL** | **CC** | **BC** | **ProCa** | **QuadCal** |
| GPT-2 Large (0.8B) | 0 | 50.25 | 56.81 | 63.22 | 60.55 | **63.70** |
| | 1 | 43.19 | 57.25 | 60.82 | 57.00 | **62.26** |
| | 4 | 46.02 | 55.80 | 64.31 | 59.58 | **66.26** |
| | 8 | 50.83 | 59.10 | 67.57 | 64.30 | **68.45** |
| GPT-2 XL (1.5B) | 0 | 46.21 | 56.12 | 62.54 | 61.12 | **64.04** |
| | 1 | 44.95 | 56.65 | 63.12 | 62.36 | **64.59** |
| | 4 | 46.31 | 57.54 | 64.83 | 62.55 | **65.19** |
| | 8 | 49.09 | 57.15 | 66.48 | 63.26 | **66.88** |
| Llama-3.2-1B-IT | 0 | 59.18 | 60.87 | **67.34** | 64.02 | 66.51 |
| | 1 | 64.88 | 64.34 | **71.11** | 69.02 | 70.94 |
| | 4 | 63.68 | 67.58 | **71.59** | 71.48 | 71.11 |
| | 8 | 66.97 | 67.93 | 72.08 | 68.81 | **72.99** |
| Llama-3.2-3B-IT | 0 | 65.69 | 67.53 | 69.94 | 67.83 | **70.58** |
| | 1 | 75.22 | 75.65 | **77.37** | 74.47 | 76.85 |
| | 4 | 74.19 | 76.69 | **79.33** | 77.66 | 78.93 |
| | 8 | 74.11 | 77.44 | **80.06** | 78.61 | 79.78 |
| Gemma-3-1B-IT | 0 | 63.77 | 62.40 | 68.17 | 64.95 | **68.55** |
| | 1 | 67.31 | 68.92 | 69.54 | 68.94 | **71.88** |
| | 4 | 65.87 | 69.07 | 68.67 | 69.27 | **72.95** |
| | 8 | 66.84 | 70.20 | 70.40 | 73.19 | **75.09** |
| Gemma-3-4B-IT | 0 | 70.97 | 69.54 | **71.15** | 70.43 | 71.04 |
| | 1 | 75.26 | 77.21 | 76.76 | **77.26** | 77.13 |
| | 4 | 76.69 | 79.45 | 78.08 | 78.89 | **79.99** |
| | 8 | 78.61 | **81.16** | 80.15 | 80.74 | 81.10 |

**erally underperforms** compared to the other calibration methods. **BC is the closest competitor to QuadCal**, making it the strongest alternative to QuadCal. Overall, these results suggest that BC is the preferred choice under confidence-based calibration methods and QuadCal is the best candidate amongst likelihood-based calibration methods by providing reliable improvements across diverse models and shot settings.

## 5.2 EFFECT OF MODEL SIZE

To assess the effect of model size on calibration, we consider the difference between the best performing calibration method (highlighted in bold) and the uncalibrated methods in Table 1. **We observe that as the model size increases, the calibration improvement decreases for instruction-tuned (IT) models.** For instance, the average performance gain after calibration using the best calibration method for the smaller IT models - Llama-3.2-1B-IT and Gemma-3-1B-IT is roughly 6-7 percentage points (pp), whereas the average performance gain for the corresponding larger IT models is only 2̃-4 pp. This clearly suggests that larger IT models benefit less from post-hoc calibration and are already better calibrated.

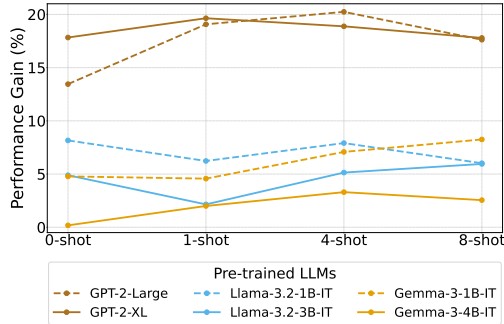

Figure 2: Performance gain (in percentage points) after calibration with the best method for each case, across ICL shot settings and pre-trained LLMs. Solid lines represent larger models, and dotted lines represent smaller models.

Interestingly, for the GPT-2 models, which are not instruction-tuned, although GPT-2 XL (1.5B) is nearly twice the size of GPT-2-Large (0.8B), calibration still provides a significant average performance gain of roughly 18 pp. Unlike the IT models, the increase in model size did not guarantee better calibration, suggesting that larger GPT-2 models are not inherently better calibrated unlike the IT models.

## 5.3 Task-level Performance

Detailed results for each model across the seven datasets under different shot settings for all calibration methods are provided in Table 2, Table 3 and Table 4, along with corresponding figures in Appendix A. Across almost all models and shot settings, **QuadCal consistently performs the best** on the **AGNews** dataset, which is a topic classification task. Similarly, on **TREC**, a question classification task, QuadCal consistently achieves the highest accuracy for GPT-2 models. For IT models, the effect varies by model size. **Smaller IT models** benefit most from likelihood-based calibration methods, especially **QuadCal**, while **larger IT models** see a stronger effect from confidence-based calibration methods, especially **CC**.

For the binary subjectivity classification dataset **Subj**, **confidence-based calibration generally performs best**, with BC frequently achieving the highest accuracy for the GPT-2 models. For Gemma models, in the 0-shot settings, confidence-based methods perform best, while in **higher shot settings, for almost all cases, QuadCal performs the best**. For Llama models, the smaller models benefit the most from BC across all shot settings, and the larger model performs better with likelihood-based calibration. This indicates that the effectiveness of the calibration depends both on the model size and the shot settings.

For **sentiment classification tasks** (SST-2, SST-5, and MR), like other tasks, the GPT-2 models benefit the most from calibration. For GPT-2, all calibration methods except CC perform competitively on SST-2, a binary sentiment dataset, **BC and ProCa generally outperform QuadCal on MR**, a binary movie review dataset, whereas on the fine-grained five-class **SST-5** dataset, **QuadCal generally performs better**. For **larger IT models**, especially on binary sentiment datasets, calibration generally provides **little to no improvement**, and when there is a gain, they are typically marginal. However, for SST-5, calibration is beneficial as the number of shots increases, indicating that additional context helps. For the smaller Llama model, likelihood-based methods generally perform best. On the other hand, for the **smaller Gemma model**, confidence-based methods are better for SST-5 and **QuadCal performs the best for SST-2** but for MR, there is no clear trend across shots or calibration methods.

Conversely, **some datasets did not benefit from calibration**. This indicates that they are either difficult to calibrate or the models are already well-calibrated for that task. This includes RTE, a textual entailment task dataset, which proves to be the most difficult to calibrate across all models and shots, as well as sentiment analysis for bigger IT models and RTE for GPT-2 models.

## 5.4 Runtime Analysis

**QuadCal is consistently faster than ProCa** across models of varying sizes, shot settings, and datasets with different number of classes. The run time for both methods increases with an increase in the number of shots and the size of the model. Figure 3 illustrates the run time comparison between QuadCal and ProCa on the TREC dataset, which has the highest number of classes, across different models and shot settings.

Interestingly, the run time difference between QuadCal and ProCa narrows as the number of shots increases. From Table 5, we can see that the average speedup is the highest for **0-shot settings**, ranging approximately from **13% to 40%**. Even in **higher-shot settings**, QuadCal maintains its efficiency, albeit smaller, ranging approximately from **1.5% to 7%**. The time taken could be further reduced by having a pre-trained QDA or GMM + Munkres model for QuadCal and ProCa respectively.

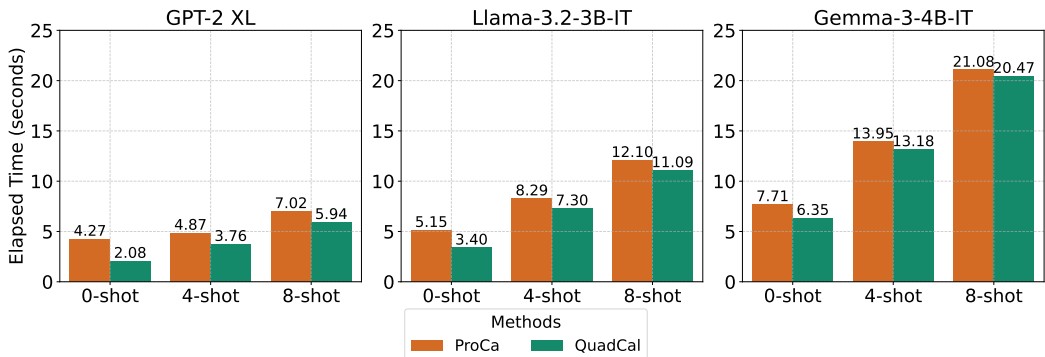

Figure 3: Computation time taken (in seconds) for QuadCal vs ProCa for the TREC dataset with six classes across different models and shot settings. QuadCal consistently shows to be faster.

### 5.5 SIGNIFICANCE TESTING

As mentioned in Section 4, we consider the results to be significant if they have higher mean accuracy or if one method performs consistently better than the other, as determined by either the **paired t-test or the binomial test**. In such a setting, we can have three types of outcomes: (i) QuadCal is significantly better than ProCa, (ii) ProCa is significantly better than QuadCal and (iii) no significant difference, where neither test indicates an advantage of one method over the other in terms of performance. This approach will ensure that we consider both the magnitude and the consistency of the performance differences between the two methods.

With six LLMs, four different shot settings and seven datasets, we evaluated a total of 168 experimental settings. Out of these, **QuadCal outperformed ProCa** in 44 cases, representing approximately **26%** of the settings. Among these, 39 cases were significant according to the binomial test, and 38 cases were significant according to the paired t-test. We observe that QuadCal shows consistently and significantly better performance than ProCa across most models, particularly on datasets such as **AG News, TREC, SST-5, and SST-2**, for various shot settings. It also performs significantly better on MR under 0-shot settings for Llama-3.2-3B-IT and Gemma-3-1B-IT, on Subj under 8-shot settings for Gemma-3-4B-IT and on RTE under 8-shot and 1-shot settings for Gemma-3-1B-IT and Gemma-3-4B-IT, respectively.

Conversely, **ProCa performed significantly better than QuadCal** for 13 cases, representing approximately **8%** of the settings. Among these, 13 cases were significant according to the binomial test, and 10 cases were significant according to the paired t-test. Interestingly, **ProCa performs significantly better than QuadCal on the Subj dataset**, especially under low-shot settings for the smaller models across all the model families. It also performs better on SST-2 under 0-shot for the GPT-2 Large model and under 1-shot and 4-shot settings for Gemma-3-4B-IT model, on MR under low-shot settings for both the GPT-2 models and the larger Gemma model, on RTE under 8-shot and 1-shot settings for the smaller GPT-2 and Llama models, respectively. Although less frequent, these results highlight that ProCa can achieve higher accuracy under certain models, shot settings and datasets.

In the remaining experimental settings, **no significant performance difference** was observed between QuadCal and ProCa, representing approximately **66%** of all cases. This suggests that for many combinations of models, shot settings, and datasets, the performance of QuadCal and ProCa is comparable.

## 6 DISCUSSION AND LIMITATIONS

**Which calibration method to choose?** As observed in Section 5, the effectiveness of a calibration method depends on several factors, including model size, model family, and the specific task or dataset. Smaller IT models and models without instruction-tuning benefit the most from calibration, whereas larger IT models benefit less from post-hoc calibration, suggesting they are already better calibrated. QuadCal consistently performs best on AGNews, TREC, and SST-5, and often

achieves higher accuracy on SST-2 and MR. This indicates that QuadCal remains effective for tasks with multiple classes that are well-distinguished and adequately represented, as confirmed by significance testing. Overall, BC and QuadCal consistently improve accuracy over uncalibrated models and frequently provide the best performance, making them reliable choices for most scenarios. While confidence-based methods may be computationally efficient, likelihood-based methods offer a Bayesian approach that is theoretically grounded and particularly suitable when reliability is critical, even if it comes with a slightly higher computational cost. However, this cost is often one-time if the task is well-defined and the calibration model is pre-trained, and among the likelihood-based calibration methods, QuadCal is up to 40% more computationally efficient than ProCa.

**Limitations**
Some of the limitations of QuadCal are inherited from ProCa and, more generally, from likelihood-based calibration methods. In particular, QuadCal requires an estimate set from the target task to train the QDA model, and assumes a fixed label space. Any change in the label space will necessitate retraining of the QDA model for calibration. QDA also becomes computationally expensive as the number of classes increases, since it requires estimating the covariance matrix and computing its inverse for each class. Alternatively, LDA could be explored for better efficiency, although it assumes the same covariance for all classes. Furthermore, like ProCa, QuadCal shifts the decision boundary and does not directly calibrate the confidence scores. This prevents the use of standard calibration assessment metrics such as ECE. Additionally, QuadCal focuses solely on mitigating contextual bias, and any inherent bias in the pre-trained LLM is left unaddressed.

## 7 CONCLUSION

We introduced QuadCal, a supervised likelihood-based calibration method for in-context learning that uses QDA to efficiently model class-conditioned distributions. Across a range of natural language classification tasks and various pre-trained LLMs, including instruction-tuned (IT) models, QuadCal matches or outperforms existing calibration methods, while being up to 40% faster than ProCa. Our results indicate that the GPT-2 models and smaller IT models benefit the most from calibration. By providing a faster Bayesian approach for calibration, QuadCal improves reliability in high-stakes domains where miscalibrated predictions could have significant consequences.

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

# A APPENDIX

Table 2: Accuracy(%) of GPT-2 models on seven text classification datasets under various ICL shot settings. Performance is reported for different calibration techniques (CC, BC, ProCa and QuadCal), ICL denotes the uncalibrated baseline. Results are the mean accuracy over 5 random seeds (mean $\pm$ standard deviation). 'Avg' and 'Med' represents macro-average and macro-median accuracy across datasets, respectively.

| Shots | Method | SST-2 | SST-5 | MR | Subj | AGNews | RTE | TREC | Avg | Med |
|---|---|---|---|---|---|---|---|---|---|---|
| | | | | | *GPT-2-Large 0.8B* | | | | | |
| 0-shot | ICL | $72.1_{0.0}$ | $26.2_{0.0}$ | $70.2_{0.0}$ | $62.1_{0.0}$ | $33.5_{0.0}$ | $53.1_{0.0}$ | $34.6_{0.0}$ | 50.25 | 53.07 |
| | CC | $80.4_{0.0}$ | $\mathbf{41.7}_{0.0}$ | $75.0_{0.0}$ | $54.5_{0.0}$ | $59.3_{0.0}$ | $\mathbf{54.9}_{0.0}$ | $32.0_{0.0}$ | 56.81 | 54.87 |
| | BC | $\mathbf{85.3}_{0.0}$ | $39.7_{0.0}$ | $81.3_{0.0}$ | $70.2_{0.0}$ | $66.5_{0.0}$ | $54.5_{0.0}$ | $45.0_{0.0}$ | 63.22 | $\mathbf{66.45}$ |
| | ProCa | $\mathbf{85.3}_{0.4}$ | $36.9_{4.7}$ | $\mathbf{82.2}_{0.5}$ | $68.7_{1.0}$ | $57.5_{3.2}$ | $52.7_{2.4}$ | $40.6_{7.3}$ | 60.55 | 57.54 |
| | QuadCal | $83.4_{0.9}$ | $40.5_{1.7}$ | $80.9_{0.8}$ | $66.2_{2.5}$ | $\mathbf{68.3}_{2.0}$ | $51.1_{3.9}$ | $\mathbf{55.5}_{3.9}$ | $\mathbf{63.70}$ | 66.19 |
| 1-shot | ICL | $56.0_{10.9}$ | $28.4_{10.4}$ | $53.3_{4.4}$ | $50.5_{1.5}$ | $28.8_{5.3}$ | $\mathbf{52.6}_{0.8}$ | $32.7_{3.4}$ | 43.19 | 50.45 |
| | CC | $75.7_{12.9}$ | $\mathbf{40.0}_{3.6}$ | $68.7_{12.1}$ | $61.4_{2.2}$ | $65.0_{5.8}$ | $50.2_{2.9}$ | $39.7_{8.1}$ | 57.25 | 61.40 |
| | BC | $\mathbf{82.9}_{1.9}$ | $38.7_{2.3}$ | $\mathbf{79.2}_{0.8}$ | $\mathbf{65.8}_{4.9}$ | $64.2_{6.4}$ | $51.5_{2.1}$ | $43.4_{4.5}$ | 60.82 | $\mathbf{64.23}$ |
| | ProCa | $\mathbf{82.9}_{2.4}$ | $31.4_{3.3}$ | $78.6_{3.1}$ | $64.7_{4.3}$ | $53.2_{8.5}$ | $50.8_{2.3}$ | $37.4_{14.3}$ | 57.00 | 53.25 |
| | QuadCal | $82.6_{2.3}$ | $38.9_{1.2}$ | $76.6_{2.4}$ | $61.6_{4.3}$ | $\mathbf{67.3}_{4.6}$ | $51.0_{2.4}$ | $\mathbf{57.7}_{5.3}$ | 62.26 | 61.62 |
| 4-shot | ICL | $52.7_{2.0}$ | $31.5_{8.5}$ | $59.4_{8.5}$ | $60.3_{10.7}$ | $33.7_{5.3}$ | $\mathbf{52.1}_{2.8}$ | $32.4_{6.9}$ | 46.02 | 52.13 |
| | CC | $70.3_{10.9}$ | $42.3_{2.3}$ | $70.1_{15.3}$ | $58.5_{10.3}$ | $56.6_{5.4}$ | $50.5_{3.6}$ | $42.3_{7.1}$ | 55.80 | 56.62 |
| | BC | $\mathbf{87.0}_{2.4}$ | $43.2_{1.6}$ | $\mathbf{81.8}_{2.5}$ | $75.8_{3.2}$ | $65.9_{7.2}$ | $51.4_{1.9}$ | $45.1_{2.5}$ | 64.31 | 65.86 |
| | ProCa | $86.3_{2.7}$ | $34.9_{1.8}$ | $79.4_{5.2}$ | $\mathbf{75.9}_{3.7}$ | $53.0_{14.6}$ | $51.8_{2.9}$ | $35.8_{8.5}$ | 59.58 | 53.03 |
| | QuadCal | $86.7_{3.5}$ | $\mathbf{43.5}_{1.2}$ | $79.2_{4.0}$ | $74.0_{3.0}$ | $\mathbf{67.3}_{4.9}$ | $48.2_{3.5}$ | $\mathbf{64.9}_{1.7}$ | $\mathbf{66.26}$ | $\mathbf{67.30}$ |
| 8-shot | ICL | $72.0_{17.4}$ | $31.2_{7.0}$ | $61.8_{9.9}$ | $57.9_{8.7}$ | $40.9_{6.7}$ | $54.1_{1.5}$ | $38.0_{6.0}$ | 50.83 | 54.13 |
| | CC | $83.8_{12.0}$ | $40.0_{5.5}$ | $72.7_{7.6}$ | $63.4_{11.0}$ | $52.6_{10.8}$ | $54.2_{1.5}$ | $47.0_{7.7}$ | 59.10 | 54.20 |
| | BC | $88.0_{3.1}$ | $\mathbf{42.7}_{3.4}$ | $83.5_{1.4}$ | $\mathbf{79.8}_{5.5}$ | $72.6_{3.0}$ | $54.0_{1.8}$ | $52.4_{3.0}$ | 67.57 | 72.60 |
| | ProCa | $88.6_{2.1}$ | $35.1_{3.9}$ | $\mathbf{83.9}_{1.5}$ | $78.6_{6.5}$ | $59.5_{12.0}$ | $\mathbf{54.3}_{0.9}$ | $50.1_{7.0}$ | 64.30 | 59.51 |
| | QuadCal | $\mathbf{89.4}_{1.1}$ | $42.1_{1.1}$ | $82.2_{2.7}$ | $78.2_{6.6}$ | $\mathbf{72.8}_{2.7}$ | $47.9_{2.9}$ | $\mathbf{66.6}_{5.2}$ | $\mathbf{68.45}$ | $\mathbf{72.78}$ |
| | | | | | *GPT-2-XL 1.5B* | | | | | |
| 0-shot | ICL | $58.6_{0.0}$ | $28.4_{0.0}$ | $58.9_{0.0}$ | $57.6_{0.0}$ | $41.5_{0.0}$ | $49.8_{0.0}$ | $28.6_{0.0}$ | 46.21 | 49.82 |
| | CC | $69.3_{0.0}$ | $22.6_{0.0}$ | $67.0_{0.0}$ | $\mathbf{72.9}_{0.0}$ | $67.7_{0.0}$ | $\mathbf{50.5}_{0.0}$ | $42.8_{0.0}$ | 56.12 | 66.98 |
| | BC | $83.6_{0.0}$ | $40.0_{0.0}$ | $80.6_{0.0}$ | $71.6_{0.0}$ | $68.1_{0.0}$ | $48.0_{0.0}$ | $45.8_{0.0}$ | 62.54 | 68.10 |
| | ProCa | $82.9_{2.5}$ | $39.0_{3.9}$ | $\mathbf{81.9}_{0.4}$ | $72.0_{1.5}$ | $59.9_{3.6}$ | $49.8_{0.5}$ | $42.4_{7.3}$ | 61.12 | 59.85 |
| | QuadCal | $\mathbf{83.7}_{1.1}$ | $\mathbf{41.6}_{2.8}$ | $81.7_{0.6}$ | $70.9_{0.8}$ | $\mathbf{68.6}_{0.4}$ | $50.1_{2.8}$ | $\mathbf{51.7}_{1.7}$ | 64.04 | 68.63 |
| 1-shot | ICL | $59.7_{14.0}$ | $26.2_{8.5}$ | $51.4_{0.6}$ | $54.4_{8.6}$ | $40.2_{10.1}$ | $\mathbf{53.6}_{0.9}$ | $29.1_{6.5}$ | 44.95 | 51.35 |
| | CC | $76.4_{2.2}$ | $30.2_{5.7}$ | $69.4_{5.0}$ | $62.0_{7.0}$ | $65.0_{3.8}$ | $52.9_{0.8}$ | $40.5_{3.4}$ | 56.65 | 62.05 |
| | BC | $83.2_{4.0}$ | $39.4_{2.4}$ | $80.1_{1.3}$ | $\mathbf{73.1}_{4.0}$ | $70.5_{3.9}$ | $49.8_{1.3}$ | $45.8_{1.5}$ | 63.12 | $\mathbf{70.55}$ |
| | ProCa | $\mathbf{90.1}_{1.5}$ | $38.4_{3.7}$ | $\mathbf{84.2}_{1.2}$ | $71.0_{5.1}$ | $67.1_{2.7}$ | $49.5_{1.8}$ | $36.4_{8.4}$ | 62.36 | 67.12 |
| | QuadCal | $86.5_{2.9}$ | $\mathbf{41.6}_{4.0}$ | $78.9_{2.6}$ | $70.5_{4.5}$ | $\mathbf{71.6}_{3.2}$ | $50.8_{4.3}$ | $\mathbf{52.2}_{4.4}$ | $\mathbf{64.59}$ | 70.55 |
| 4-shot | ICL | $66.3_{13.7}$ | $31.4_{7.4}$ | $56.5_{5.9}$ | $53.4_{5.0}$ | $40.9_{13.0}$ | $52.0_{3.3}$ | $23.8_{5.7}$ | 46.31 | 51.99 |
| | CC | $79.9_{10.2}$ | $33.5_{3.5}$ | $67.7_{8.9}$ | $68.0_{8.7}$ | $59.9_{6.4}$ | $\mathbf{52.8}_{0.6}$ | $41.1_{4.5}$ | 57.54 | 59.92 |
| | BC | $90.1_{0.8}$ | $40.7_{3.0}$ | $77.3_{11.4}$ | $74.1_{10.0}$ | $72.8_{5.6}$ | $51.4_{1.7}$ | $47.3_{3.6}$ | 64.83 | 72.84 |
| | ProCa | $89.8_{0.8}$ | $35.1_{5.0}$ | $\mathbf{78.3}_{11.9}$ | $\mathbf{74.3}_{9.8}$ | $68.7_{8.0}$ | $51.5_{1.5}$ | $40.2_{7.5}$ | 62.55 | 68.66 |
| | QuadCal | $\mathbf{90.3}_{0.8}$ | $\mathbf{42.8}_{1.5}$ | $76.1_{11.2}$ | $73.3_{11.2}$ | $\mathbf{74.1}_{4.9}$ | $47.9_{3.2}$ | $\mathbf{51.9}_{1.8}$ | $\mathbf{65.19}$ | $\mathbf{73.31}$ |
| 8-shot | ICL | $57.0_{9.0}$ | $30.6_{7.9}$ | $65.2_{12.7}$ | $57.9_{11.3}$ | $42.9_{4.2}$ | $52.9_{2.2}$ | $37.2_{5.0}$ | 49.09 | 52.90 |
| | CC | $73.9_{11.5}$ | $28.7_{3.4}$ | $74.2_{8.3}$ | $68.3_{8.3}$ | $55.9_{14.0}$ | $\mathbf{53.2}_{0.3}$ | $45.9_{1.7}$ | 57.15 | 55.92 |
| | BC | $87.3_{1.9}$ | $39.2_{2.8}$ | $\mathbf{80.7}_{5.8}$ | $\mathbf{79.9}_{3.2}$ | $76.3_{3.5}$ | $51.5_{1.3}$ | $50.6_{2.9}$ | 66.48 | 76.27 |
| | ProCa | $\mathbf{87.5}_{1.9}$ | $36.4_{4.0}$ | $80.4_{6.1}$ | $78.4_{3.1}$ | $69.4_{7.7}$ | $51.4_{1.7}$ | $39.5_{6.6}$ | 63.26 | 69.38 |
| | QuadCal | $86.5_{2.5}$ | $\mathbf{43.4}_{1.2}$ | $77.0_{6.9}$ | $79.4_{3.4}$ | $\mathbf{78.6}_{3.0}$ | $49.0_{4.2}$ | $\mathbf{54.4}_{4.6}$ | $\mathbf{66.88}$ | $\mathbf{77.02}$ |

## Usage of LLMs

The free version of ChatGPT was primarily used to refine and polish the text, which was originally written by the authors. No text generated by ChatGPT was directly included. It was also used for coding tasks, particularly for visualizations. The code to draw the ellipse in Figure 1 was generated by ChatGPT. For other plots, either the authors wrote the initial draft of the code and refined it with ChatGPT, or ChatGPT provided the initial draft which was then refined by the authors. Essentially, it was used as an alternative to search engines for coding tasks.

Table 3: Accuracy(%) of Llama models on seven text classification datasets under various ICL shot settings. Performance is reported for different calibration techniques (CC, BC, ProCa and QuadCal), ICL denotes the uncalibrated baseline. Results are the mean accuracy over 5 random seeds (mean ± standard deviation). 'Avg' and 'Med' represents macro-average and macro-median accuracy across datasets, respectively.

| Shots | Method | SST-2 | SST-5 | MR | Subj | AGNews | RTE | TREC | Avg | Median |
|---|---|---|---|---|---|---|---|---|---|---|
| | | | | | *Llama-3.2-IT 1B* | | | | | |
| 0-shot | ICL | $86.7_{0.0}$ | $38.5_{0.0}$ | $84.2_{0.0}$ | $62.4_{0.0}$ | $47.6_{0.0}$ | $57.0_{0.0}$ | $37.8_{0.0}$ | 59.18 | 57.04 |
| | CC | $89.7_{0.0}$ | $\mathbf{46.6}_{0.0}$ | $85.2_{0.0}$ | $53.6_{0.0}$ | $61.6_{0.0}$ | $49.1_{0.0}$ | $40.2_{0.0}$ | 60.87 | 53.60 |
| | BC | $89.1_{0.0}$ | $41.7_{0.0}$ | $85.7_{0.0}$ | $\mathbf{63.2}_{0.0}$ | $69.1_{0.0}$ | $\mathbf{66.4}_{0.0}$ | $\mathbf{56.2}_{0.0}$ | **67.34** | **66.43** |
| | ProCa | $\mathbf{90.2}_{0.6}$ | $41.4_{2.2}$ | $\mathbf{86.5}_{0.5}$ | $63.2_{0.1}$ | $56.3_{8.6}$ | $65.9_{0.6}$ | $44.7_{6.4}$ | 64.02 | 63.16 |
| | QuadCal | $89.1_{1.0}$ | $45.3_{3.5}$ | $86.2_{0.3}$ | $59.6_{1.1}$ | $\mathbf{69.9}_{3.2}$ | $60.9_{5.4}$ | $54.5_{2.0}$ | 66.51 | 60.94 |
| 1-shot | ICL | $88.7_{3.5}$ | $41.2_{5.4}$ | $84.0_{3.7}$ | $60.3_{5.8}$ | $76.5_{3.9}$ | $50.6_{1.7}$ | $52.8_{4.8}$ | 64.88 | 60.28 |
| | CC | $85.7_{7.6}$ | $35.4_{6.8}$ | $84.7_{4.1}$ | $63.5_{2.5}$ | $78.2_{7.7}$ | $49.4_{1.4}$ | $53.5_{7.3}$ | 64.34 | 63.52 |
| | BC | $90.4_{2.1}$ | $43.1_{2.7}$ | $85.6_{2.3}$ | $\mathbf{69.2}_{4.1}$ | $84.0_{1.1}$ | $66.1_{1.8}$ | $59.4_{1.6}$ | **71.11** | **69.17** |
| | ProCa | $89.3_{2.2}$ | $39.2_{6.3}$ | $86.2_{2.3}$ | $68.8_{4.3}$ | $82.7_{1.7}$ | $\mathbf{67.2}_{1.8}$ | $49.8_{8.7}$ | 69.02 | 68.83 |
| | QuadCal | $\mathbf{91.8}_{0.9}$ | $\mathbf{44.5}_{1.6}$ | $\mathbf{86.8}_{1.5}$ | $63.9_{3.3}$ | $\mathbf{84.2}_{0.9}$ | $61.7_{3.0}$ | $\mathbf{63.7}_{2.6}$ | 70.94 | 63.88 |
| 4-shot | ICL | $92.9_{0.8}$ | $42.2_{6.5}$ | $83.3_{4.7}$ | $59.1_{7.7}$ | $70.3_{11.7}$ | $49.3_{2.0}$ | $48.6_{7.1}$ | 63.68 | 59.10 |
| | CC | $93.2_{1.4}$ | $38.9_{5.2}$ | $86.5_{1.8}$ | $69.7_{10.0}$ | $80.0_{4.2}$ | $51.7_{5.9}$ | $53.1_{13.9}$ | 67.58 | 69.65 |
| | BC | $93.4_{0.3}$ | $42.4_{2.7}$ | $86.5_{2.3}$ | $\mathbf{73.9}_{7.5}$ | $80.2_{5.2}$ | $63.4_{3.0}$ | $61.2_{5.4}$ | **71.59** | **73.90** |
| | ProCa | $91.1_{2.7}$ | $38.6_{6.6}$ | $\mathbf{87.5}_{1.1}$ | $72.8_{7.3}$ | $80.7_{2.4}$ | $\mathbf{64.9}_{2.3}$ | $\mathbf{64.6}_{6.1}$ | 71.48 | 72.85 |
| | QuadCal | $\mathbf{93.7}_{0.3}$ | $\mathbf{44.9}_{4.7}$ | $87.3_{2.2}$ | $72.5_{9.0}$ | $\mathbf{83.1}_{1.0}$ | $62.7_{1.4}$ | $53.6_{26.3}$ | 71.11 | 72.51 |
| 8-shot | ICL | $92.5_{2.0}$ | $45.5_{3.9}$ | $87.1_{2.9}$ | $54.6_{2.9}$ | $80.4_{4.7}$ | $52.2_{6.0}$ | $56.3_{11.6}$ | 66.97 | 56.32 |
| | CC | $92.8_{1.1}$ | $39.9_{4.4}$ | $\mathbf{89.0}_{1.0}$ | $64.4_{7.4}$ | $79.2_{5.7}$ | $51.2_{6.7}$ | $59.0_{8.7}$ | 67.93 | 64.40 |
| | BC | $93.0_{1.3}$ | $42.4_{3.1}$ | $88.5_{1.4}$ | $\mathbf{70.6}_{4.6}$ | $83.0_{1.9}$ | $64.5_{2.7}$ | $62.6_{4.4}$ | 72.08 | **70.57** |
| | ProCa | $90.1_{2.3}$ | $36.0_{8.6}$ | $87.5_{1.4}$ | $67.3_{10.7}$ | $82.3_{2.4}$ | $65.0_{2.9}$ | $53.5_{11.0}$ | 68.81 | 67.29 |
| | QuadCal | $\mathbf{93.2}_{0.9}$ | $\mathbf{45.9}_{1.9}$ | $88.3_{1.5}$ | $68.1_{6.5}$ | $\mathbf{83.7}_{1.8}$ | $63.2_{5.6}$ | $\mathbf{68.6}_{3.3}$ | **72.99** | 68.64 |
| | | | | | *Llama-3.2-IT 3B* | | | | | |
| 0-shot | ICL | $\mathbf{91.2}_{0.0}$ | $\mathbf{48.1}_{0.0}$ | $87.2_{0.0}$ | $49.4_{0.0}$ | $53.0_{0.0}$ | $\mathbf{75.1}_{0.0}$ | $55.8_{0.0}$ | 65.69 | 55.80 |
| | CC | $89.2_{0.0}$ | $48.0_{0.0}$ | $84.2_{0.0}$ | $49.4_{0.0}$ | $73.2_{0.0}$ | $72.9_{0.0}$ | $55.8_{0.0}$ | 67.53 | 72.92 |
| | BC | $91.0_{0.0}$ | $44.2_{0.0}$ | $\mathbf{87.3}_{0.0}$ | $49.6_{0.0}$ | $77.7_{0.0}$ | $75.1_{0.0}$ | $64.6_{0.0}$ | 69.94 | **75.09** |
| | ProCa | $90.4_{0.8}$ | $36.5_{3.6}$ | $86.0_{0.9}$ | $\mathbf{50.0}_{0.5}$ | $73.3_{4.8}$ | $72.3_{2.7}$ | $66.3_{5.2}$ | 67.83 | 72.35 |
| | QuadCal | $91.0_{0.2}$ | $46.7_{2.1}$ | $\mathbf{87.3}_{0.2}$ | $49.9_{0.5}$ | $78.7_{1.0}$ | $71.5_{0.9}$ | $\mathbf{69.0}_{6.7}$ | 70.58 | 71.48 |
| 1-shot | ICL | $93.8_{1.3}$ | $46.7_{1.3}$ | $89.2_{1.2}$ | $72.7_{5.5}$ | $84.1_{1.3}$ | $76.1_{2.0}$ | $63.9_{5.7}$ | 75.22 | 76.10 |
| | CC | $92.2_{2.9}$ | $46.1_{2.3}$ | $87.2_{2.4}$ | $69.8_{4.6}$ | $83.2_{3.0}$ | $75.3_{1.7}$ | $\mathbf{75.7}_{4.1}$ | 75.65 | 75.72 |
| | BC | $94.1_{1.0}$ | $46.8_{1.0}$ | $89.6_{1.0}$ | $\mathbf{77.3}_{2.8}$ | $85.3_{0.8}$ | $\mathbf{77.3}_{1.8}$ | $71.4_{4.3}$ | **77.37** | **77.26** |
| | ProCa | $93.2_{1.6}$ | $41.7_{5.8}$ | $\mathbf{89.7}_{0.8}$ | $75.3_{3.3}$ | $84.1_{1.5}$ | $76.5_{2.2}$ | $60.8_{5.8}$ | 74.47 | 76.46 |
| | QuadCal | $\mathbf{94.4}_{0.9}$ | $\mathbf{47.8}_{3.5}$ | $88.7_{2.1}$ | $74.8_{3.3}$ | $\mathbf{85.5}_{1.0}$ | $74.6_{2.3}$ | $72.2_{5.1}$ | 76.85 | 74.75 |
| 4-shot | ICL | $\mathbf{95.7}_{0.3}$ | $45.5_{2.1}$ | $\mathbf{90.5}_{0.8}$ | $55.8_{3.0}$ | $82.7_{2.3}$ | $78.8_{2.5}$ | $70.3_{5.0}$ | 74.19 | 78.84 |
| | CC | $95.3_{0.6}$ | $39.8_{3.2}$ | $89.0_{1.7}$ | $77.7_{8.9}$ | $84.4_{2.0}$ | $77.3_{3.6}$ | $73.3_{4.2}$ | 76.69 | 77.70 |
| | BC | $\mathbf{95.7}_{0.3}$ | $45.8_{1.6}$ | $90.5_{0.8}$ | $81.7_{3.7}$ | $84.4_{1.1}$ | $\mathbf{80.6}_{1.5}$ | $\mathbf{76.6}_{3.2}$ | **79.33** | 81.73 |
| | ProCa | $95.2_{0.3}$ | $39.0_{2.6}$ | $89.8_{2.1}$ | $82.8_{3.5}$ | $84.1_{0.8}$ | $79.2_{3.5}$ | $73.6_{6.8}$ | 77.66 | 82.84 |
| | QuadCal | $95.5_{0.4}$ | $\mathbf{48.0}_{1.6}$ | $90.1_{0.8}$ | $\mathbf{83.6}_{4.0}$ | $85.2_{0.6}$ | $79.8_{2.0}$ | $70.3_{7.0}$ | 78.93 | 83.59 |
| 8-shot | ICL | $\mathbf{95.9}_{0.3}$ | $45.6_{2.1}$ | $90.5_{1.3}$ | $53.5_{2.4}$ | $82.5_{2.4}$ | $78.9_{4.0}$ | $71.8_{3.0}$ | 74.11 | 78.91 |
| | CC | $95.4_{0.5}$ | $37.1_{3.4}$ | $89.1_{2.0}$ | $81.4_{5.3}$ | $84.3_{1.9}$ | $77.0_{7.9}$ | $\mathbf{77.8}_{1.9}$ | 77.44 | 81.42 |
| | BC | $95.8_{0.3}$ | $46.5_{2.5}$ | $\mathbf{91.1}_{0.5}$ | $85.1_{4.1}$ | $84.9_{0.9}$ | $80.4_{1.9}$ | $76.6_{1.3}$ | **80.06** | 84.94 |
| | ProCa | $94.7_{1.1}$ | $46.2_{6.7}$ | $89.8_{1.4}$ | $86.0_{1.9}$ | $84.4_{1.7}$ | $\mathbf{80.9}_{1.0}$ | $68.4_{5.7}$ | 78.61 | 84.38 |
| | QuadCal | $95.5_{0.4}$ | $\mathbf{48.8}_{1.8}$ | $90.7_{0.5}$ | $\mathbf{87.3}_{3.6}$ | $85.0_{1.1}$ | $79.7_{2.2}$ | $71.3_{3.6}$ | 79.78 | **85.05** |

Table 4: Accuracy(%) of Gemma models on seven text classification datasets under various ICL shot settings. Performance is reported for different calibration techniques (CC, BC, ProCa and QuadCal), ICL denotes the uncalibrated baseline. Results are the mean accuracy over 5 random seeds (mean ± standard deviation). 'Avg' and 'Med' represents macro-average and macro-median accuracy across datasets, respectively.

| Shots | Method | SST-2 | SST-5 | MR | Subj | AGNews | RTE | TREC | Avg | Median |
|---|---|---|---|---|---|---|---|---|---|---|
| | | | | | *Gemma-3-IT 1B* | | | | | |
| 0-shot | ICL | $86.7_{0.0}$ | $39.5_{0.0}$ | $82.8_{0.0}$ | $61.8_{0.0}$ | $37.0_{0.0}$ | $68.6_{0.0}$ | $70.0_{0.0}$ | 63.77 | 68.59 |
| | CC | $82.7_{0.0}$ | $\mathbf{42.5}_{0.0}$ | $78.4_{0.0}$ | $\mathbf{62.4}_{0.0}$ | $42.9_{0.0}$ | $67.9_{0.0}$ | $60.0_{0.0}$ | 62.40 | 62.40 |
| | BC | $86.9_{0.0}$ | $40.4_{0.0}$ | $\mathbf{83.6}_{0.0}$ | $62.4_{0.0}$ | $65.7_{0.0}$ | $\mathbf{69.3}_{0.0}$ | $69.0_{0.0}$ | 68.17 | 69.00 |
| | ProCa | $84.0_{1.2}$ | $37.2_{2.2}$ | $79.6_{1.2}$ | $62.3_{0.4}$ | $57.0_{2.4}$ | $67.2_{2.0}$ | $67.4_{2.7}$ | 64.95 | 67.22 |
| | QuadCal | $\mathbf{87.2}_{0.5}$ | $41.5_{1.5}$ | $82.1_{0.7}$ | $59.6_{0.4}$ | $\mathbf{70.3}_{2.9}$ | $66.8_{1.6}$ | $\mathbf{72.4}_{2.8}$ | **68.55** | **70.31** |
| 1-shot | ICL | $89.8_{1.8}$ | $45.0_{0.7}$ | $83.8_{0.6}$ | $53.7_{1.6}$ | $75.2_{2.7}$ | $61.7_{1.5}$ | $62.0_{4.9}$ | 67.31 | 62.00 |
| | CC | $90.1_{2.3}$ | $\mathbf{46.0}_{0.9}$ | $84.6_{0.8}$ | $60.5_{8.2}$ | $75.1_{3.9}$ | $60.4_{1.7}$ | $65.8_{3.7}$ | 68.92 | 65.76 |
| | BC | $90.0_{1.7}$ | $44.6_{1.6}$ | $84.1_{0.4}$ | $61.7_{4.5}$ | $77.4_{2.4}$ | $63.8_{1.2}$ | $65.2_{4.1}$ | 69.54 | 65.24 |
| | ProCa | $90.0_{1.2}$ | $39.7_{3.8}$ | $\mathbf{85.1}_{1.4}$ | $63.4_{5.8}$ | $77.5_{2.2}$ | $63.0_{1.6}$ | $63.8_{2.6}$ | 68.94 | 63.80 |
| | QuadCal | $\mathbf{90.5}_{1.3}$ | $43.7_{2.5}$ | $84.7_{0.7}$ | $\mathbf{64.6}_{6.7}$ | $\mathbf{82.0}_{1.2}$ | $\mathbf{66.1}_{2.7}$ | $\mathbf{71.7}_{5.0}$ | **71.88** | **71.68** |
| 4-shot | ICL | $89.7_{3.5}$ | $\mathbf{45.0}_{1.9}$ | $85.7_{0.6}$ | $61.0_{8.5}$ | $70.7_{5.5}$ | $60.6_{3.5}$ | $48.5_{12.6}$ | 65.87 | 60.95 |
| | CC | $91.0_{1.6}$ | $44.8_{2.3}$ | $\mathbf{86.4}_{0.5}$ | $68.6_{6.5}$ | $73.4_{3.0}$ | $59.9_{3.1}$ | $59.4_{7.3}$ | 69.07 | 68.57 |
| | BC | $90.1_{3.0}$ | $44.9_{1.3}$ | $85.9_{0.6}$ | $71.4_{6.9}$ | $74.2_{3.7}$ | $62.5_{3.6}$ | $51.7_{11.5}$ | 68.67 | 71.41 |
| | ProCa | $91.1_{1.5}$ | $41.6_{3.5}$ | $85.6_{0.3}$ | $73.4_{7.6}$ | $73.0_{2.6}$ | $63.5_{2.6}$ | $56.7_{7.2}$ | 69.27 | 72.95 |
| | QuadCal | $\mathbf{91.5}_{0.9}$ | $44.7_{3.6}$ | $85.7_{0.6}$ | $\mathbf{73.7}_{6.3}$ | $\mathbf{80.0}_{2.3}$ | $\mathbf{64.9}_{5.5}$ | $\mathbf{70.2}_{4.4}$ | **72.95** | **73.71** |
| 8-shot | ICL | $90.2_{1.5}$ | $44.3_{2.4}$ | $84.1_{2.5}$ | $62.2_{6.3}$ | $81.4_{1.1}$ | $60.9_{2.7}$ | $44.9_{10.4}$ | 66.84 | 62.18 |
| | CC | $90.4_{2.8}$ | $43.6_{3.9}$ | $84.7_{3.8}$ | $78.4_{7.3}$ | $78.5_{1.9}$ | $60.6_{2.4}$ | $55.3_{6.6}$ | 70.20 | 78.37 |
| | BC | $90.4_{1.4}$ | $\mathbf{44.9}_{2.2}$ | $84.7_{1.9}$ | $77.9_{3.0}$ | $81.5_{0.8}$ | $63.5_{2.6}$ | $50.0_{6.9}$ | 70.40 | 77.88 |
| | ProCa | $91.3_{1.7}$ | $42.8_{5.4}$ | $\mathbf{86.6}_{0.8}$ | $82.0_{1.8}$ | $80.4_{1.4}$ | $65.8_{1.2}$ | $63.4_{4.6}$ | 73.19 | 80.43 |
| | QuadCal | $\mathbf{92.0}_{0.9}$ | $44.4_{2.4}$ | $\mathbf{86.6}_{0.5}$ | $\mathbf{82.1}_{2.5}$ | $\mathbf{81.7}_{2.1}$ | $\mathbf{68.1}_{1.4}$ | $70.7_{2.5}$ | **75.09** | **81.71** |
| | | | | | *Gemma-3-IT 4B* | | | | | |
| 0-shot | ICL | $90.3_{0.0}$ | $\mathbf{45.6}_{0.0}$ | $86.5_{0.0}$ | $50.0_{0.0}$ | $80.0_{0.0}$ | $74.0_{0.0}$ | $70.4_{0.0}$ | 70.97 | 74.01 |
| | CC | $\mathbf{91.8}_{0.0}$ | $30.3_{0.0}$ | $\mathbf{88.1}_{0.0}$ | $50.0_{0.0}$ | $80.5_{0.0}$ | $74.7_{0.0}$ | $\mathbf{71.4}_{0.0}$ | 69.54 | 74.73 |
| | BC | $90.9_{0.0}$ | $43.9_{0.0}$ | $87.2_{0.0}$ | $\mathbf{50.5}_{0.0}$ | $80.5_{0.0}$ | $74.0_{0.0}$ | $70.4_{0.0}$ | **71.15** | 74.73 |
| | ProCa | $91.4_{1.4}$ | $40.9_{2.9}$ | $87.8_{0.3}$ | $49.9_{0.8}$ | $81.1_{0.6}$ | $73.7_{1.9}$ | $68.4_{0.8}$ | 70.43 | 73.65 |
| | QuadCal | $91.8_{1.3}$ | $44.8_{1.6}$ | $86.5_{1.1}$ | $49.9_{0.4}$ | $\mathbf{82.0}_{0.8}$ | $\mathbf{76.4}_{0.8}$ | $66.1_{5.4}$ | 71.04 | **76.39** |
| 1-shot | ICL | $\mathbf{95.9}_{0.3}$ | $50.0_{2.9}$ | $90.6_{1.0}$ | $59.9_{5.0}$ | $80.7_{2.2}$ | $75.9_{1.4}$ | $74.0_{0.5}$ | 75.26 | 75.88 |
| | CC | $95.8_{0.3}$ | $\mathbf{50.8}_{2.5}$ | $90.8_{0.9}$ | $73.6_{12.2}$ | $78.0_{3.9}$ | $76.5_{1.1}$ | $\mathbf{74.9}_{1.5}$ | 77.21 | 76.53 |
| | BC | $\mathbf{95.9}_{0.2}$ | $50.3_{2.8}$ | $90.6_{0.9}$ | $69.3_{4.6}$ | $81.2_{1.9}$ | $76.1_{1.4}$ | $74.0_{0.6}$ | 76.76 | 76.10 |
| | ProCa | $95.8_{0.2}$ | $48.9_{4.2}$ | $\mathbf{90.8}_{0.8}$ | $75.2_{7.8}$ | $81.6_{2.0}$ | $76.1_{1.5}$ | $72.3_{1.9}$ | **77.26** | 76.10 |
| | QuadCal | $95.2_{0.8}$ | $48.2_{4.9}$ | $90.2_{1.6}$ | $72.1_{6.7}$ | $\mathbf{84.1}_{1.5}$ | $\mathbf{78.3}_{1.0}$ | $71.8_{2.8}$ | 77.13 | **78.34** |
| 4-shot | ICL | $96.0_{0.4}$ | $47.7_{1.1}$ | $\mathbf{91.4}_{1.1}$ | $69.3_{10.1}$ | $80.8_{4.1}$ | $80.6_{2.0}$ | $71.2_{6.4}$ | 76.69 | 80.58 |
| | CC | $96.0_{0.5}$ | $47.0_{5.0}$ | $91.3_{1.0}$ | $82.8_{3.2}$ | $80.9_{2.6}$ | $\mathbf{80.7}_{2.0}$ | $\mathbf{77.5}_{4.9}$ | 79.45 | 80.90 |
| | BC | $96.0_{0.4}$ | $48.7_{1.0}$ | $91.3_{1.1}$ | $76.2_{6.5}$ | $81.9_{3.1}$ | $80.6_{2.0}$ | $71.8_{6.5}$ | 78.08 | 80.58 |
| | ProCa | $\mathbf{96.1}_{0.2}$ | $44.9_{4.3}$ | $91.3_{0.5}$ | $82.7_{3.4}$ | $83.5_{1.5}$ | $80.5_{2.0}$ | $73.2_{4.5}$ | 78.89 | 82.66 |
| | QuadCal | $95.2_{0.6}$ | $\mathbf{50.3}_{2.5}$ | $91.3_{0.4}$ | $\mathbf{83.0}_{3.3}$ | $\mathbf{84.4}_{1.2}$ | $80.5_{2.2}$ | $75.1_{8.2}$ | **79.99** | **83.01** |
| 8-shot | ICL | $95.4_{0.9}$ | $49.1_{1.5}$ | $90.9_{2.1}$ | $72.5_{10.5}$ | $85.6_{1.5}$ | $80.7_{1.8}$ | $76.0_{3.1}$ | 78.61 | 80.65 |
| | CC | $95.5_{0.4}$ | $48.0_{6.0}$ | $91.3_{1.5}$ | $86.8_{3.8}$ | $\mathbf{86.8}_{0.3}$ | $\mathbf{81.4}_{2.2}$ | $\mathbf{78.4}_{3.2}$ | **81.16** | **86.80** |
| | BC | $95.5_{0.8}$ | $\mathbf{51.5}_{0.7}$ | $91.0_{1.9}$ | $82.0_{4.8}$ | $85.8_{1.2}$ | $80.5_{1.6}$ | $74.8_{3.0}$ | 80.15 | 81.99 |
| | ProCa | $\mathbf{95.6}_{0.5}$ | $46.9_{5.3}$ | $\mathbf{91.3}_{0.8}$ | $87.2_{3.3}$ | $85.9_{0.7}$ | $81.0_{2.0}$ | $77.2_{4.7}$ | 80.74 | 85.92 |
| | QuadCal | $95.4_{0.2}$ | $49.0_{3.2}$ | $\mathbf{91.3}_{0.5}$ | $\mathbf{89.0}_{2.4}$ | $85.0_{1.2}$ | $80.7_{2.2}$ | $77.2_{5.4}$ | 81.10 | 85.04 |

Table 5: Computation time (in seconds) of QuadCal and ProCa across different models, shot settings, and datasets. The average speedup (%) across datasets highlights the efficiency of QuadCal, particularly in low-shot settings.

| Model | Shots | SST-2 | | AGNews | | TREC | | Avg speedup (%) |
|---|---|---|---|---|---|---|---|---|
| | | ProCa | QuadCal | ProCa | QuadCal | ProCa | QuadCal | |
| GPT-2 XL | 0 | 2.98 | 1.91 | 4.56 | 3.01 | 4.27 | 2.08 | **40.4%** |
| | 4 | 5.34 | 5.10 | 9.25 | 8.51 | 4.87 | 3.76 | 11.8% |
| | 8 | 9.01 | 8.81 | 16.44 | 15.81 | 7.02 | 5.94 | 7.1% |
| Llama-3.2-3B-IT | 0 | 4.06 | 3.02 | 7.03 | 5.42 | 5.15 | 3.40 | **27.5%** |
| | 4 | 10.08 | 9.75 | 16.49 | 16.04 | 8.29 | 7.30 | 6.0% |
| | 8 | 16.64 | 16.33 | 29.92 | 29.77 | 12.10 | 11.09 | 3.6% |
| Gemma-3-4B-IT | 0 | 6.49 | 5.61 | 11.38 | 10.33 | 7.71 | 6.35 | **13.5%** |
| | 4 | 17.63 | 17.52 | 29.85 | 29.58 | 13.95 | 13.18 | 2.3% |
| | 8 | 29.13 | 28.82 | 54.92 | 54.45 | 21.08 | 20.47 | 1.6% |

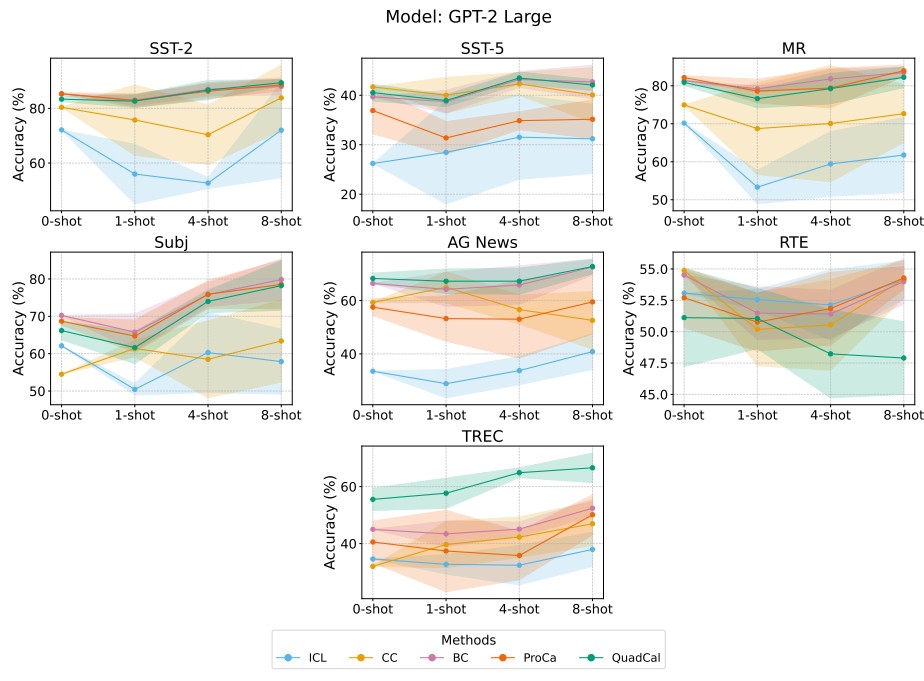

Figure 4: Accuracy(%) of the GPT-2-Large (0.8B) model across [0, 1, 4, 8]-shot settings for seven natural language classification datasets. The four different calibration methods are compared against the uncalibrated ICL baseline.

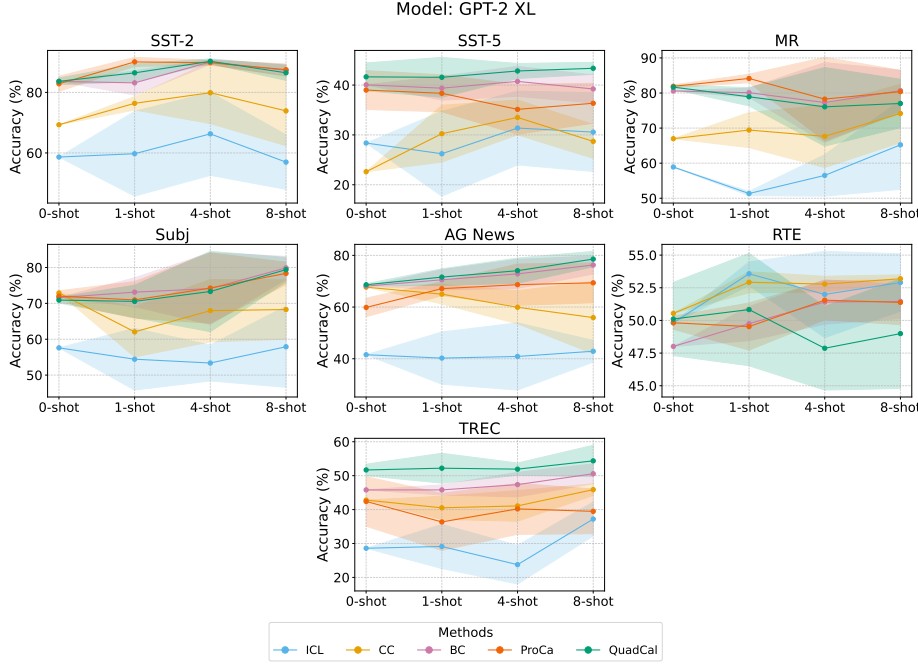

Figure 5: Accuracy(%) of the GPT-2-XL (1.5B) model across [0, 1, 4, 8]-shot settings for seven natural language classification datasets. The four different calibration methods are compared against the uncalibrated ICL baseline.

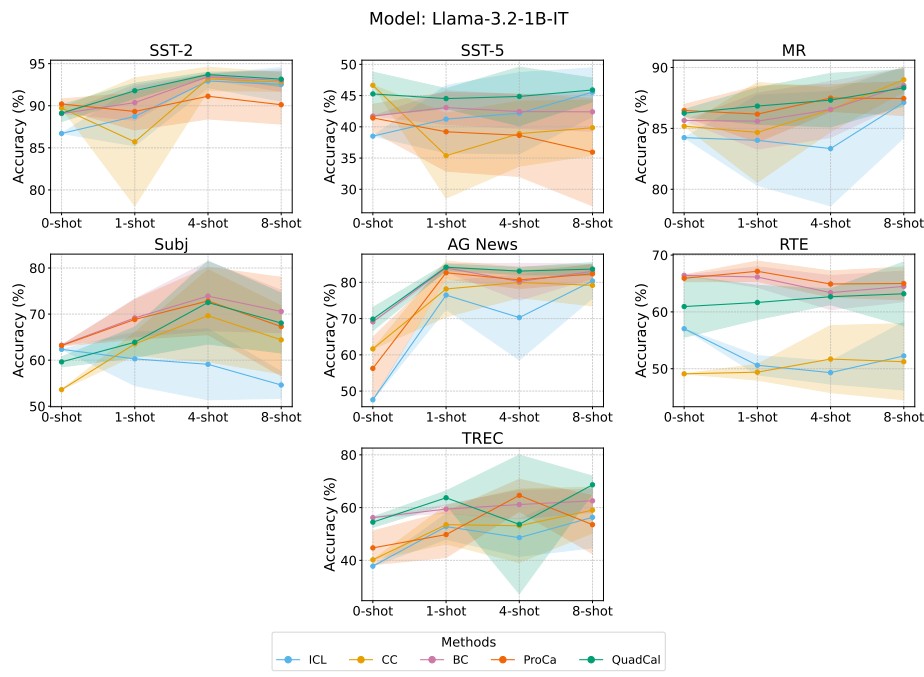

Figure 6: Accuracy(%) of the Llama-3.2-IT 1B model across [0, 1, 4, 8]-shot settings for seven natural language classification datasets. The four different calibration methods are compared against the uncalibrated ICL baseline.

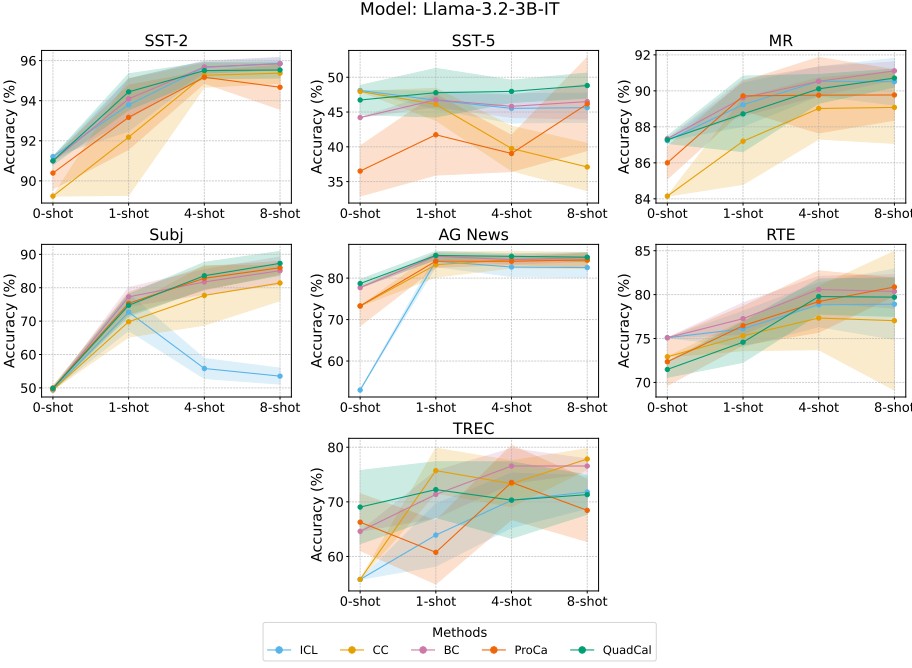

Figure 7: Accuracy(%) of the Llama-3.2-IT 3B model across [0, 1, 4, 8]-shot settings for seven natural language classification datasets. The four different calibration methods are compared against the uncalibrated ICL baseline.

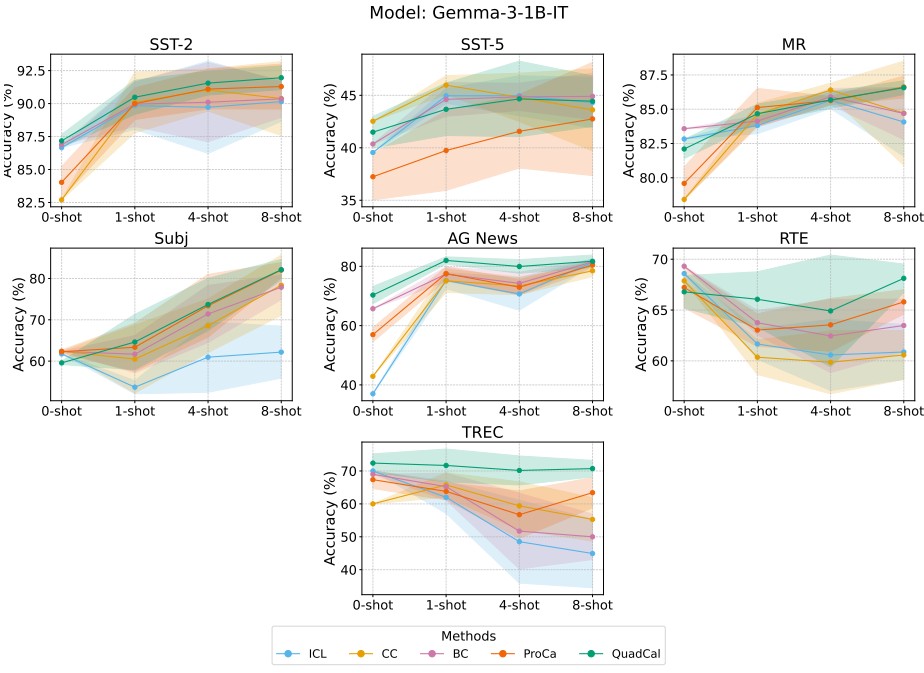

Figure 8: Accuracy(%) of the Gemma-3-IT 1B model across [0, 1, 4, 8]-shot settings for seven natural language classification datasets. The four different calibration methods are compared against the uncalibrated ICL baseline.

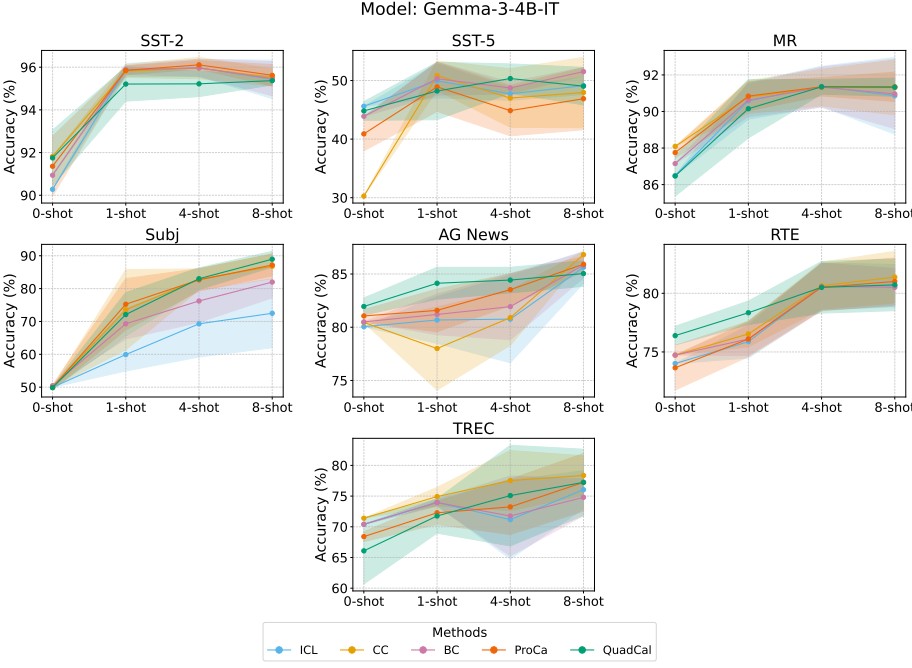

Figure 9: Accuracy(%) of the Gemma-3-IT 4B model across [0, 1, 4, 8]-shot settings for seven natural language classification datasets. The four different calibration methods are compared against the uncalibrated ICL baseline.

