# OpenReview forum: "QuadCal: Calibration for In-Context Learning"
_ICLR.cc/2026/Conference — ICLR 2026 Conference Withdrawn Submission_

### Official Review · Reviewer_GCzA · 2025-10-25

**Soundness:** 2
**Presentation:** 1
**Contribution:** 2
**Rating:** 2
**Confidence:** 4

**Summary:**

In-context learning is widely applied to high-stakes domains with high consequences for errors. Existing calibration methods may  introduce contextual bias or introduce unnecessary computational overhead. The paper proposed a supervised likelihood-based calibration method that is up to 40% faster and outperforms the existing likelihood-based approach. Experiments across classification datasets valiate the efficiency of the proposed method.

**Strengths:**

1. The paper introduces a novel supervised likelihood-based calibration method which achieve low computational cost and higher macro-average accuracy.

2. The methodology is easy to understand and follow.

3. Comprehensive empirical analysis and ablation studies in the experiment section.

**Weaknesses:**

1. Tables 1 show that the results of QuadCal and the baselines are quite similar, especially for the advanced LLMs like Llama-3.2-1B or 3B. It would be helpful if the authors could further clarify the advantages of the proposed methods.

2. The paper should compare QuadCal with the advanced and recent baseline methods from other studies, such as LPC[3] and SupICL[4].  Such an analysis could offer valuable insights into the strengths and weaknesses of the proposed approaches.

3. The author should complete the full 9 pages. The figure and its illustration should be shown on the same page, such as Figure 1. Section 3.1 should be titled Motivation or Methodology rather than Background.

4. The paper validate the proposed method only on simple classification datasets. The authors should add more complex downstream tasks such as question-answering tasks or code generation tasks.

5. The authors should report the results of the Expected Calibration Error (ECE), Maximum Calibration Error (MCE), Adaptive ECE, and Brier Score, which are widely used to evaluate the performance of calibration methods.

6. The absence of code makes it to reproduce the results claimed in the paper or verify the method's effectiveness on the tasks.

7. It would be useful to report the performance of in-context learning using a calibration method based on Linear Discriminant Analysis.

8. It would be useful to compare the computational cost with more baselines such as CC and BC.

9. Which dataset is used in Figure 1?

10. Missing the details of used dataset, such as dataset sizes, prompts.

11. Missing analysis on the selection strategy used in ICL: It would be useful if the authors also studied how different selection strategies (e.g., EPR [1] and DPP [2]).

12. Missing the analysis of different prompt formation.

13. Missing the analysis of larger model sizes (such as 7B, 14B, 30 B and 70B) and API (such as GPT-4o, Claude and Deepseek).

[1] Learning to retrieve prompts for in-context learning.

[2] Compositional exemplars for in-context learning

[3] Enhancing In-context Learning via Linear Probe Calibration

[4] Large Language Models are Miscalibrated In-Context Learners

**Questions:**

See Weaknesses

---

> ### Author Response · Authors · 2025-11-25
> **Response to Reviewer GCzA**
>
> We would like to thank the reviewer for their feedback and for acknowledging the strengths of our work.
>
> [Q1:] Tables 1 show that the results of QuadCal and the baselines are quite similar, especially for the advanced LLMs like Llama-3.2-1B or 3B. It would be helpful if the authors could further clarify the advantages of the proposed methods.
>
> [A1:] Please see (1.) from the recurring reviewer comments
>
> [Q2:]The paper should compare QuadCal with the advanced and recent baseline methods from other studies, such as LPC[3] and SupICL[4]. Such an analysis could offer valuable insights into the strengths and weaknesses of the proposed approaches.
>
> [A2:] Thank you for this suggestion. We will take this into consideration.
>
> [Q3:] The author should complete the full 9 pages. The figure and its illustration should be shown on the same page, such as Figure 1. Section 3.1 should be titled Motivation or Methodology rather than Background.
>
> [A3:] We thank the reviewers for their feedback, we will re-position Section 3 accordingly, and as we incorporate all the feedback.
>
> [Q4:] The paper validate the proposed method only on simple classification datasets. The authors should add more complex downstream tasks such as question-answering tasks or code generation tasks.
>
> [A4:] Please see (2.) from the recurring reviewer comments
>
> [Q5:] The authors should report the results of the Expected Calibration Error (ECE), Maximum Calibration Error (MCE), Adaptive ECE, and Brier Score, which are widely used to evaluate the performance of calibration methods.
>
> [A5:] The likelihood-based calibration methods do not tweak the confidence scores directly. This inherent limitation prevents the use of calibration assessment metrics to assess performance before and after calibration. We note this in lines 448-450 in the Limitations section.
>
> [Q6:] The absence of code makes it to reproduce the results claimed in the paper or verify the method's effectiveness on the tasks.
>
> [A6:] We planned to share the code via the ICLR discussion forum, which we understood would allow reviewers to access it. We now realize that the forum is only open during the review phase. The code will be made open-source to ensure reproducibility.
>
> [Q7:] It would be useful to report the performance of in-context learning using a calibration method based on Linear Discriminant Analysis.
>
> [A7:] Please see (3.) from the recurring reviewer comments
>
> [Q8:] It would be useful to compare the computational cost with more baselines such as CC and BC.
>
> [A8:] Since the confidence-based methods (CC, BC) do not require any model training, it will definitely be faster than any of the likelihood-based approaches. The run time analysis was mainly done to support our theoretical claim that QuadCal would be faster because it is a 1-step process whereas the existing method is a 2-step process adding more compute time.
>
> [Q9:] Which dataset is used in Figure 1?
>
> [A9:] We generated the dataset from two different Gaussian distributions. The intent of the plot is to illustrate how QDA can effectively separate classes with distinct covariances.
>
> [Q10:] Missing the details of used dataset, such as dataset sizes, prompts.
>
> [A10:] Thank you for the feedback, we will add more information in our Appendix section
>
> [Q11:] Missing analysis on the selection strategy used in ICL: It would be useful if the authors also studied how different selection strategies (e.g., EPR [1] and DPP [2]).
>
> [A11:] We appreciate the reviewer’s suggestion, we currently do random sampling as done in all the previous works.
>
> [Q12:] Missing the analysis of different prompt formation.
>
> [A12:] Yes, we are aware of this and we plan to expand our work to include this in future experiments.
>
> [Q13:] Missing the analysis of larger model sizes (such as 7B, 14B, 30 B and 70B) and API (such as GPT-4o, Claude and Deepseek).
>
> [A13:] Our initial focus was on recent models in comparable size to the GPT models used in ProCa for benchmarking. While we recognize the value of analysis on larger models, we will explore it within our computational constraints. However, our focus will remain on open-sourced models due to budget constraints.

---

> > ### Comment · Reviewer_GCzA · 2025-11-26
> >
> > The authors did not address my questions with experiments. Therefore, I maintain my score.

---

### Official Review · Reviewer_xqkx · 2025-11-01

**Soundness:** 2
**Presentation:** 3
**Contribution:** 2
**Rating:** 2
**Confidence:** 3

**Summary:**

This paper introduces a supervised likelihood-based calibration method for large language models in-context learning.  The whole frame work is built on ProCa but replacing Gaussian Mixture Model with Quadratic Discriminant Analysis.  Experiments across multiple models, e.g. GPT-2, Llama and Gemma and different classification tasks demonstrate its performance.  The results indicate that smaller or non-instruction-tuned models particularly benefit more from this post-hoc  likelihood-based calibration process.

**Strengths:**

1. The paper is well-written and easy to follow.
2. The experiments are fairly designed, including multiple model families and various task domains, as well as different in-context learning shot settings.
3. The main claim, such as faster runtime and improved average performance, is supported by experimental results and comparisons.
4. The discussions section is good regarding when different calibration methods, either confidence-based or likelihood-based, are preferable.

**Weaknesses:**

1. The observation that larger size or instruction-tuned language models tend to be better calibrated is not new and has been reported in prior work, such as “_Language Models (Mostly) Know What They Know._” So, findings in this paper mainly confirm existing understanding rather than providing new insights.
2. The inclusion of statistical testing is good, but it seems unnecessary to me.  Moreover, the paper does not present detailed test statistics and full testing reports, even though the authors discussed and compared the results and significance levels.
3.  Expect for the reduction in computational cost, the improved accuracy of the proposed approach is not very significant for the majority of tasks, which limits its practical advantage.
4. Although positioned as a theoretically grounded Bayesian approach, this work does not clearly articulate how or why its likelihood-based formulation provides deeper theoretical justification compared to confidence-based methods like BC, which achieves comparable performance with greater implementation efficiency.
5. The construction of the estimate set is not well explained. And all of the methodological explanation relies only on a toy example (Figure 1), which makes the implementation details unclear.

**Questions:**

Please see the weakness.

---

> ### Author Response · Authors · 2025-11-25
> **Response to Reviewer xqkx**
>
> We would like to thank the reviewer for their feedback and for acknowledging the strengths of our work.
>
> [Q1] The observation that larger size or instruction-tuned language models tend to be better calibrated is not new and has been reported in prior work, such as “Language Models (Mostly) Know What They Know.” So, findings in this paper mainly confirm existing understanding rather than providing new insights.
>
> [A1] Thank you for pointing it out. In lines 214-215, we note that IT models are difficult to calibrate and cited a relevant paper. Regarding model size, we agree; to avoid miscommunication, we will remove the bold formatting and add proper citation. However, the major contribution of the paper remains that we propose a supervised, likelihood-based calibration method that is more theoretically motivated and is computationally faster than the existing method.
>
> [Q2] The inclusion of statistical testing is good, but it seems unnecessary to me. Moreover, the paper does not present detailed test statistics and full testing reports, even though the authors discussed and compared the results and significance levels.
>
> [A2] We believe that statistical testing is essential to support any claim and to prove that the observations are not due to random chance. We have described the setup (tests, significance level) in Section 4 and the results in Section 5.5. Does the reviewer refer to the p-value for each of the tests? It would be helpful if the reviewer could clarify on this, we will add them in the Appendix section.
>
> [Q3] Expect for the reduction in computational cost, the improved accuracy of the proposed approach is not very significant for the majority of tasks, which limits its practical advantage.
>
> [A3] Please see (1.) from the recurring reviewer comments
>
> [Q4] Although positioned as a theoretically grounded Bayesian approach, this work does not clearly articulate how or why its likelihood-based formulation provides deeper theoretical justification compared to confidence-based methods like BC, which achieves comparable performance with greater implementation efficiency.
>
> [A4] Please see (4.) from the recurring reviewer comments
>
> [Q5] The construction of the estimate set is not well explained. And all of the methodological explanation relies only on a toy example (Figure 1), which makes the implementation details unclear.
>
> [A5] We briefly describe the construction of the estimate set between lines 224 and 231, and we will improve readability.

---

### Official Review · Reviewer_PMDq · 2025-11-01

**Soundness:** 2
**Presentation:** 2
**Contribution:** 2
**Rating:** 2
**Confidence:** 4

**Summary:**

This paper introduces QuadCal, a supervised likelihood-based calibration method for in-context learning (ICL) that addresses contextual bias in large language models. The key innovation is replacing Prototypical Calibration (ProCa)'s unsupervised GMM clustering + Hungarian algorithm approach with Quadratic Discriminant Analysis (QDA), which directly models class-conditioned distributions using ground-truth labels. The authors evaluate QuadCal against existing calibration methods (CC, BC, ProCa) on seven NLP classification datasets using GPT-2 models and instruction-tuned Llama/Gemma models across 0/1/4/8-shot settings.

**Strengths:**

1. While the core idea of substituting GMM with QDA is straightforward, the application to ICL calibration is sensible and previously unexplored.
2. The key insight, that supervised learning eliminates the need for expensive cluster-to-label mapping, is valid and practically useful.
3. The paper clearly motivates the problem (contextual bias in ICL, computational overhead in ProCa) and positions QuadCal as a solution.
4. For practitioners deploying LLMs in high-stakes domains, this work offers a useful, faster alternative for likelihood-based calibration.

**Weaknesses:**

1. In Table 1, the performance gain seems to be trivial for the majority of the models. In fact, BC outperforms QuadCal a few times. QuadCal significantly outperforms ProCa in only 26% of 168 settings, where 66% show no significant difference, and ProCa is significantly better in 8% of cases. For a paper whose main claim is improved performance, these statistics are weak and are insufficient to claim a meaningful contribution.
2. The datasets used are pretty saturated and could already have been covered by some models' pre-training and post-training.  It is generally a good practice to test on ICL tasks that are new, hard, and require the model to learn novel logic or strategies, since the value of ICL is to learn new tasks at inference time. For example, tasks like BBH, BB-extra-Hard, ZebraLogic, GPQA, MMLU-Pro, and other logical puzzles or algorithmic reasoning challenges should be the main evaluation focus. The current set of datasets does not support a generalized claim and undermines the impact of the contribution.
3. There are some potential missed baselines. There is no comparison with Linear Discriminant Analysis, which would be faster than QDA, and test whether class-specific covariances are necessary. Also, there are no comparisons with simpler supervised methods like logistic regression on log-probabilities or ensemble approaches.
4. The discussion notes that different methods work better for different tasks but provides no principled explanation:
- Why does QuadCal excel on AGNews and TREC but not RTE?
- What properties of these datasets drive the differences?
- No analysis of class balance, class separability, or output probability distributions
5. The core contribution, replacing GMM+Munkres with QDA, is primarily an engineering substitution rather than a fundamental methodological advance. While the paper claims QDA avoids ProCa's computational overhead, the theoretical justification is superficial. Why should QDA outperform GMM for this specific problem? Under what data distributions or class structures would QDA be preferred? The statement that ProCa's restriction to n clusters makes GMM "functionally similar" to QDA (lines 183-186) needs rigorous proof. GMM with n components can still capture multimodal within-class distributions, while QDA assumes unimodality.

**Questions:**

1. The observation that larger IT models benefit less from calibration is interesting but underdeveloped:
- Are larger models inherently better calibrated? Are there ECE comparisons on the model sizes?
- Is this specific to instruction-tuning or general to scale?

---

> ### Author Response · Authors · 2025-11-25
> **Response to Reviewer PMDq**
>
> We would like to thank the reviewer for their feedback and for acknowledging the strengths of our work.
>
> We request the reviewer to see the recurring reviewer comments where we address all the questions posed in the Weaknesses section.

---

### Author Response · Authors · 2025-11-25
**Recurring reviewer comments**

We would like to thank all the reviewers for their helpful feedback to make the work better. We would like to address some recurring reviewer comments here:

(1.) Performance gain-
While it is true that QuadCal shows modest improvements, the main contributions of our work are reducing the computational overhead and providing a theoretically motivated, supervised likelihood-based calibration method. Given that QuadCal significantly outperforms ProCa in 26% of the 168 settings and shows no significant difference in 66% of the cases, we can still opt for QuadCal in 92% of cases due to its computational efficiency and more reliable supervised approach. These advantages would help in practical settings where we have to deploy it on edge devices and for real-time calibration.

(2.) Datasets-
We retained the datasets used in earlier works to enable benchmarking. Many of the tasks and datasets suggested by the reviewers (reasoning, QA or generation tasks) do not have a fixed label space, which is essential for likelihood-based calibration method (as noted in our Limitations section) and hence we would not be able to use those datasets. However, we have identified a few datasets that we plan to include in the near future.

(3.) LDA-
We plan to include results from LDA in the paper soon. We excluded ensemble methods since we expect them to be prone to overfitting.

(4.) Theoretical justification-
We will strengthen the theoretical justification by constructing failure cases to better motivate the use of QDA.

---

### Author Response · Authors · 2025-11-26

Dear Reviewers:
Thank you for your feedback, we would like to refine our work based on your feedback and due to time constraints, we would like to withdraw our submission.

---

### Note · Authors · 2025-11-26

I have read and agree with the venue's withdrawal policy on behalf of myself and my co-authors.